# Phosphatidylinositol 3-kinase and COPII generate LC3 lipidation vesicles from the ER-Golgi intermediate compartment

**Liang Ge, Min Zhang, Randy Schekman\***

Department of Molecular and Cell Biology, Howard Hughes Medical Institute, University of California, Berkeley, Berkeley, United States

**Abstract** Formation of the autophagosome requires significant membrane input from cellular organelles. However, no direct evidence has been developed to link autophagic factors and the mobilization of membranes to generate the phagophore. Previously, we established a cell-free LC3 lipidation reaction to identify the ER-Golgi intermediate compartment (ERGIC) as a membrane source for LC3 lipidation, a key step of autophagosome biogenesis (Ge et al., eLife 2013; 2:e00947). We now report that starvation activation of autophagic phosphotidylinositol-3 kinase (PI3K) induces the generation of small vesicles active in LC3 lipidation. Subcellular fractionation studies identified the ERGIC as the donor membrane in the generation of small lipidation-active vesicles. COPII proteins are recruited to the ERGIC membrane in starved cells, dependent on active PI3K. We conclude that starvation activates the autophagic PI3K, which in turn induces the recruitment of COPII to the ERGIC to bud LC3 lipidation-active vesicles as one potential membrane source of the autophagosome.

**\*For correspondence:**
schekman@berkeley.edu

**Reviewing editor**: Noboru Mizushima, The University of Tokyo, Japan

## Main text

Macroautophagy (hereafter autophagy) is a conserved cellular process wherein cytoplasmic protein and organelles are packaged and delivered to the lysosome to promote survival under conditions of stress (*Choi et al., 2013*; *Jiang and Mizushima, 2014*; *Mizushima et al., 2008*). The process begins with the formation of a double-membrane organelle, termed the autophagosome, which envelops part of the cytoplasm for targeting to and degradation in the lysosome (*Burman and Ktistakis, 2010*; *Weidberg et al., 2011*; *Hamasaki et al., 2013b*; *Lamb et al., 2013*; *Ge et al., 2014*). Completion of the autophagosome requires step-wise acquisition of membranes from intracellular organelles directed by proteins devoted to autophagy (*Mizushima et al., 2011*; *Rubinsztein et al., 2012*; *Abada and Elazar, 2014*; *Feng et al., 2014*). How an endomembrane organelle responds to an autophagic signal to generate autophagosomal precursors is unclear.

A direct link between the autophagic signal and the biogenesis of the phagophore membrane is the autophagic phosphotidylinositol-3 kinase (PI3K) complex (VPS34, VPS15, Beclin-1 and ATG14)-mediated production of phosphotidylinositol-3 phosphate (PI3P), an event triggered by starvation (*Sun et al., 2008*; *Matsunaga et al., 2010*; *Obara and Ohsumi, 2011*). We have previously established a cell-free LC3 lipidation reaction that is dependent on PI3K and its lipid product, PI3P (*Ge and Schekman, 2013*; *Ge et al., 2013*). In *Atg5* knockout (KO) mouse embryonic fibroblasts (MEF), which are deficient in the terminal step of the LC3 lipidation cascade, autophagosome formation is blocked downstream of the PI3K pathway (*Mizushima et al., 2001*; *Suzuki et al., 2007*; *Itakura and Mizushima, 2010*). Therefore, membrane precursors acting between the PI3K pathway and phagophore maturation may accumulate in *Atg5* KO MEFs after starvation.

To study the PI3K-induced early event, we employed the lipidation assay to compare the sensitivity to PI3K inhibition between membranes from untreated and starved *Atg5* KO MEFs (*Figure 1A*). Consistent with the previous study, lipidation of LC3 on the untreated membrane was efficiently

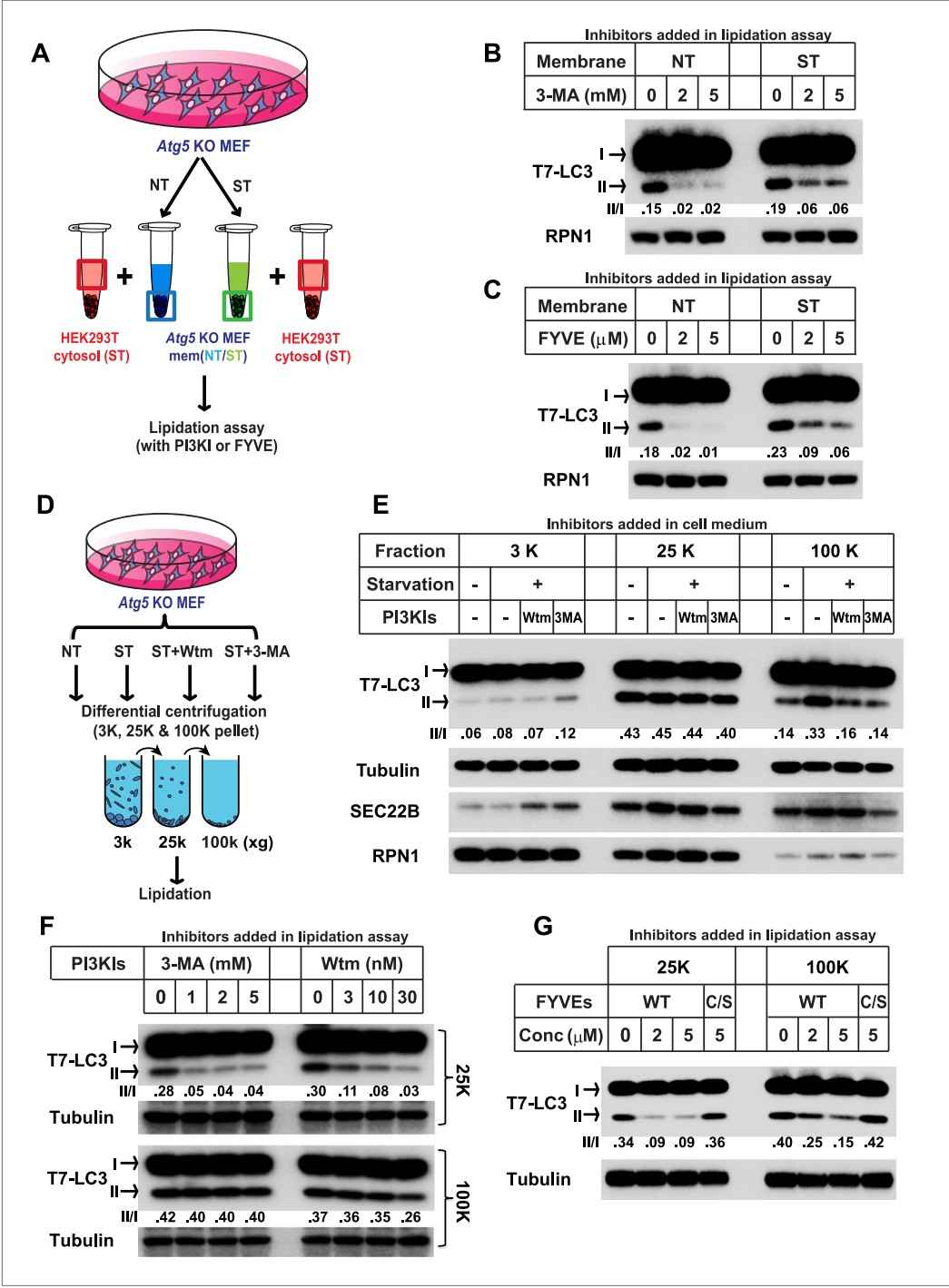

**Figure 1**. Starvation and PI3K-dependent generation of small membranes for LC3 lipidation. (**A**–**C**) *Atg5* KO MEFs were either untreated (NT) or starved (ST) with EBSS (Earle's Balanced Salt Solution) for 30 min. Total membranes (mem) from lysed cells were collected and incubated in a lipidation reaction with cytosols prepared from starved HEK293T cells. Reactions contained the indicated concentrations of PI3K inhibitor (PI3KI) 3-methyladenine (3-MA) (**B**) or FYVE protein (**C**). A diagram of the experimental scheme is shown in (**A**). RPN1, Ribophorin 1 (**D**, **E**) *Atg5* KO MEFs were either untreated (NT) or starved (ST) with EBSS in the absence or presence of 20 nM wortmannin (Wtm) or 10 mM 3-methyladenine (3-MA) for 30 min. Membranes from each treated cell sample were collected and subjected to a differential centrifugation to separate the 3K ×*g*, 25K ×*g* and 100K ×*g* pellet fractions followed by a lipidation assay as above (**E**). A diagram is shown in (**D**). (**F**, **G**) *Atg5* KO MEFs were starved for 30 min. Membranes in the 25K ×*g* and 100K ×*g* pellets from a differential centrifugation were collected as described above. A similar
*Figure 1. Continued on next page*

*Figure 1. Continued*

lipidation assay was performed in the presence of indicated concentrations (Conc in **G**) of 3-MA, wortmannin (**F**) and FYVE protein as well as a PI3P binding-deficient FYVE mutant protein (C/S) (**G**). Quantification of lipidation activity is shown as the ratio of LC3-II to LC3-I (II/I).
The following figure supplement is available for figure 1:

**Figure supplement 1**. The FYVE domain protein blocks LC3 lipidation of the 25K membrane pellet fraction.

---

blocked by a PI3K inhibitor 3-methyladenine (3-MA, ~sevenfold decrease of activity with the indicated concentration of 3-MA, *Figure 1B*) or the PI3P blocker FYVE domain protein (~ninefold and 18-fold decrease of activity with the indicated concentration of FYVE protein, *Figure 1C*) (*Stenmark and Aasland, 1999*; *Axe et al., 2008*). However, LC3 lipidation promoted with membranes from starved cells was less sensitive to 3-MA or FYVE domain protein inhibition (~threefold decrease with the indicated concentration of 3-MA, *Figure 1B*, and ~twofold and fourfold decrease with indicated concentration of FYVE domain protein, *Figure 1C*), indicating that a later autophagosomal precursor, bypassing the need of PI3K for LC3 lipidation, was generated in response to starvation in *Atg5* KO MEFs.

To separate the precursor membranes active in LC3 lipidation as well as to determine the requirement of PI3K in generating them, we took membrane samples of untreated or starved *Atg5* KO MEFs incubated with or without PI3K inhibitors. A differential centrifugation protocol similar to that described in our previous study (*Ge et al., 2013*) was performed with lysed cell preparations followed by incubation of membranes under conditions that promote the lipidation of LC3 (*Figure 1D*). Consistent with the previous result (*Ge et al., 2013*), the 25K membrane from untreated cells had the highest activity whereas neither the 3K nor the 100K membrane pellet fractions had comparable activity (~1/7 and 1/3 of the activity of the 25K membrane in the 3K and 100K membrane respectively, *Figure 1E*). Starvation or PI3K inhibition did not substantially affect the lipidation activity in the 3K or 25K fractions (*Figure 1E*). However, the 100K membrane from starved cells, mainly containing small vesicles, displayed a ~1.5-fold increase of lipidation activity compared to that of membranes from untreated cells (*Figure 1E*). Addition of PI3K inhibitors to cells during starvation abolished the increase of lipidation activity in the 100K membrane induced by starvation (*Figure 1E*). Therefore, starvation induces the formation of small membranes competent for LC3 lipidation in a process that requires activation of the PI3K.

To test if the membrane precursors generated by starvation is a downstream effect of PI3K activation, we performed the lipidation assay with the 100K membrane and compared its sensitivity to PI3K inhibition as well as PI3P occlusion with the 25K membrane (*Figure 1F*) which contains the ER-Golgi intermediate compartment that induces LC3 lipidation in a PI3K-dependent manner (*Ge et al., 2013*). Consistent with our previous result, lipidation activity in the 25K membranes was inhibited by PI3K inhibitors 3-MA and wortmannin as well as the PI3P blocker, FYVE domain protein, in a dose dependent manner (*Figure 1F,G* and *Figure 1—figure supplement 1*). In contrast, PI3K inhibitors did not inhibit lipidation activity of the 100K membranes (*Figure 1F*). Likewise, the FYVE domain protein was less potent in inhibiting LC3 lipidation with membranes in the 100K than the 25K pellet fraction (*Figure 1G*). Nonetheless, membranes in the 100K pellet fraction remained partially sensitive to the FYVE domain protein suggesting that access of PI3P in these membranes remained critical for the recruitment of factors, such as WIPI2 (*Dooley et al., 2014*), necessary for the lipidation event. These data suggest that membranes in the 100K fraction are generated downstream of the PI3K-dependent step.

To determine if the small membranes were generated from a big donor organelle, we employed a cell-free small membrane formation reaction based on our previous work on the cell-free formation of COPII vesicles (*Bednarek et al., 1995*; *Kim et al., 2005*). A medium-speed membrane pellet fraction from lysed *Atg5* KO MEFs was collected and incubated with *Atg5* KO MEF cytosol together with GTP and an ATP regeneration system to generate small vesicles which were concentrated by a two-step centrifugation (*Figure 2A*). The LC3 lipidation reaction was performed on the slowly-sedimenting vesicles (*Figure 2A*). As shown in *Figure 2B*, COPII vesicles, marked by the reporter protein SEC22B, were generated in the presence of cytosol, membrane and nucleotide (*Figure 2B*). Starvation signals from

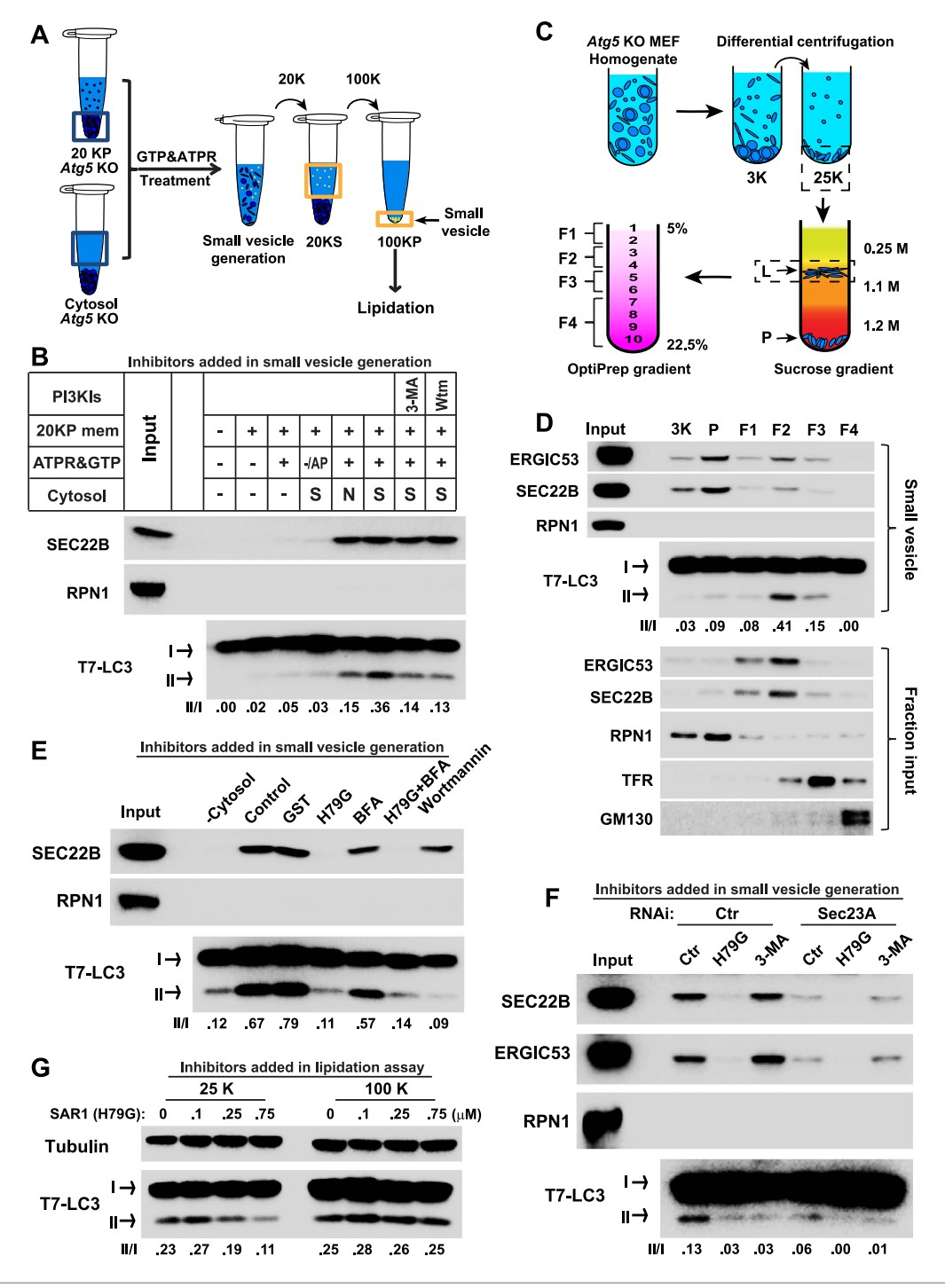

**Figure 2**. COPII and PI3K-dependent generation of small vesicles from ERGIC for LC3 lipidation. (**A**) Diagram showing the cell-free system to generate small vesicles for LC3 lipidation. Briefly, a medium-speed membrane pellet (20K ×g, 20KP) from *Atg5* KO MEFs was harvested and incubated with cytosol from *Atg5* KO MEFs, GTP and an ATP regeneration system (ATPR), and in the absence or presence of PI3K inhibitors (**Figure 2B**) or the PI3P blocker FYVE domain protein (**Figure 2—figure supplement 1**). The slowly-sedimenting membranes generated were separated from the donor membrane by a medium speed centrifugation (20K ×g) and collected by high-speed sedimentation (100K ×g) of the supernatant fraction (20KS). A lipidation assay with HEK293T cytosol was performed to analyze the competency of the small membranes (100KP) to induce LC3 lipidation. (**B**) A small vesicle generation *Figure 2. Continued on next page*

*Figure 2. Continued*

assay as above was performed with the indicated conditions. Slowly-sedimenting vesicles were collected to determine activity in the LC3 lipidation reaction. 3-MA, 5 mM; Wortmannin, 20 nM; S, starved *Atg5* KO MEF cytosol; N, untreated *Atg5* KO MEF cytosol; −/AP, in the absence of ATP regeneration and GTP and in the presence of apyrase (AP). (**C**) Diagram showing the fractionation procedure and fractions collected for the small vesicle generation assay shown in (**D**). Briefly, *Atg5* KO MEF membranes were subjected to differential centrifugation to collect 3K ×*g* and 25K ×*g* pellet fractions. Sucrose gradient ultracentrifugation was performed to separate the L and P fraction after which the L was further separated by OptiPrep gradient ultracentrifugtion. Ten fractions from top to bottom were collected and combined to four as indicated (F1-4). (**D**) A small vesicle generation assay followed by a lipidation reaction was performed with the 3K, P and F1-4 fractions (upper four panels: Small vesicle). SDS-PAGE and immunoblot with indicated antibodies were carried out to probe the organelle enrichment of each fraction (lower five panels: Fraction input). TFR, transferrin receptor (**E**) A small vesicle generation assay as shown in (**A**) was performed in the presence of indicated proteins or drugs. Slowly-sedimenting vesicles were collected and incubated in a lipidation reaction in the presence of GST, H79G (Sar1A (H79G), 0.7 µM), wortmannin (20 nM), or BFA (brefeldin A, 0.5 µg/ml). (**F**) A small vesicle generation assay as shown in (**A**) was performed with cytosols from starved *Atg5* KO MEF from control or Sec23A knockdown cells in the absence or presence of Sar1A (H79G) or 3-MA followed by a lipidation reaction. (**G**) *Atg5* KO MEFs were starved for 30 min. Differential centrifugation as shown in **Figure 1D** was performed to collect the 25K ×*g* and 100K ×*g* pellet fractions. The LC3 lipidation was performed on these two fractions with indicated concentrations of SAR1 (H79G). Quantification of lipidation activity is shown as the ratio of LC3-II to LC3-I (II/I).

The following figure supplements are available for figure 2:

**Figure supplement 1**. PI3P is required for generation of the LC3 lipidation-active vesicles in vitro.

**Figure supplement 2**. Lipidation on the small vesicles generated in vitro is resistant to PI3K inhibition.

**Figure supplement 3**. Early autophagic factors are required for the generation of small vesicles active in LC3 lipidation.

**Figure supplement 4**. Immunodepletion of the ERGIC abolishes the generation of LC3 lipidation-active vesicles.

**Figure supplement 5**. Knockdown efficiency of the Sec23A siRNA.

**Figure supplement 6**. PI3K and COPII-dependent generation of small vesicles positive for endogenous LC3 with WT MEF cytosol and membrane.

**Figure supplement 7**. Knockdown of Sec23A inhibits autophagosome biogenesis.

**Figure supplement 8**. Small LC3 vesicles colocalize with COPII after starvation COS7.

the cytosol or inhibition of PI3K activity did not affect the level of budded SEC22B (**Figure 2B**). Slowly-sedimenting vesicles generated in this reaction supported LC3 lipidation with a ~1.5-fold increase in lipidation activity in incubations containing cytosol from starved compared to untreated cells (**Figure 2B**). This stimulatory effect was compromised by addition of PI3K inhibitors during the small vesicle formation step (**Figure 2B**). Occlusion of PI3P by FYVE domain protein also attenuated the generation of small vesicles active in LC3 lipidation (**Figure 2—figure supplement 1**), consistent with the requirement for PI3P production on the donor membrane to generate the vesicles active in the lipidation of LC3. Importantly, lipidation activity promoted by the small vesicles was resistant to PI3K inhibition but remained partly sensitive to the PI3P blocker peptide, which was similar to the behavior of the 100K membrane generated from starved cells (**Figure 2—figure supplement 2**). The data indicate that a donor membrane generates these small vesicles in a starvation-induced and PI3K-dependent manner.

During the induction of autophagy, the serine/threonine complex containing FIP200, ULK1, ATG13 and ATG101 is activated and precedes the recruitment of the autophagic PI3K complex consisting of ATG14, Beclin-1, VPS34 and VPS15 to the target membrane. These events lead to the generation of PI3P on the local membrane to initiate autophagosome biogenesis (**Burman and Ktistakis, 2010**;

*Mizushima et al., 2011*; *Rubinsztein et al., 2012*; *Abada and Elazar, 2014*; *Feng et al., 2014*). To test the requirement of the upstream autophagic factors involved in the generation of LC3 lipidation-active vesicles, we used RNAi to deplete FIP200, ATG14 and Beclin-1, key components of the upstream autophagic complexes, in *Atg5* KO MEFs (*Figure 2—figure supplement 3*). Cytosol deficient in any one of these proteins reduced the efficiency with which LC3 lipidation-active vesicles were generated in the cell-free assay (*Figure 2—figure supplement 3*). Therefore the data suggest that the production of the lipidation-active vesicles is regulated by upstream autophagic factors.

To probe the membrane source responsible for generating the small vesicles, we employed a three-step membrane fractionation approach (*Ge et al., 2013*) and incubated separated membrane fractions in the cell-free small vesicle formation assay (*Figure 2C*). The membrane template for COPII budding, marked by SEC22B and ERGIC53 (two COPII cargos generated from ER and enriched in ERGIC; *Hauri et al., 2000*; *Mancias and Goldberg, 2007*; *Zhang et al., 1999*), was enriched in the P fraction, where the majority of the endoplasmic reticulum (ER) sedimented (the enrichment of RPN1, an ER marker, *Figure 2D*). We observed that the ERGIC fraction (F2) was most active in generating small vesicles active in LC3 lipidation, although it was less active in capturing cargo proteins into COPII vesicles than membranes in the P fraction (*Figure 2D*). The plasma membrane and endosome fraction (marked by transferrin receptor), and the Golgi fraction (marked by GM130) were less potent in the generation of both COPII and lipidation-active vesicles (*Figure 2D*).

To confirm the role of the ERGIC in the production of vesicles active in LC3 lipidation, immuno-depletion of the ERGIC from the L fraction was performed with SEC22B antibody as described previously (*Ge et al., 2013*). Depletion of the ERGIC membrane decreased the generation of small vesicles marked by COPII cargos (ERGIC53 and Sec22B) and active in LC3 lipidation (*Figure 2—figure supplement 4*), consistent with a role for the ERGIC as the donor membrane in generating lipidation-active vesicles.

COPII and COPI coats have been functionally associated with the ERGIC (*Appenzeller-Herzog and Hauri, 2006*; *Zanetti et al., 2012*; *Brandizzi and Barlowe, 2013*). To test if COPII and COPI are involved in the generation of lipidation active vesicles, we used a COPII inhibitor SAR1 (H79G) and a COPI inhibitor brefeldin A (BFA) (*Peyroche et al., 1999*; *Ward et al., 2001*) in the cell-free small vesicle formation assay (*Figure 2E*). SAR1 (H79G) inhibited COPII budding and the generation of lipidation-active vesicles whereas BFA showed a marginal effect (*Figure 2E*). Consistent with this, we found that cytosol collected from *Sec23A*-silenced *Atg5* KO MEFs (*Figure 2F* and *Figure 2—figure supplement 5A*) was less active in the generation of small vesicles competent as a template for lipidation of LC3 (*Figure 2F*). To further characterize the role of COPII, we collected the 25K membrane pellet fraction, containing the donor membrane ERGIC, and the 100K membrane pellet fraction, enriched in autophagosomal precursors, from starved *Atg5* KO MEF and then performed the LC3 lipidation assay in the presence of the SAR1 mutant (H79G). Inhibition of COPII decreased LC3 lipidation in the 25K fraction but did not affect LC3 lipidation on the 100K fraction (*Figure 2G*). This result is consistent with the view that COPII acts on the ERGIC membrane to generate the lipidation-active membranes, i.e. the small vesicles are downstream products of COPII budding.

The above experiments were performed in autophagy-deficient *Atg5* KO MEF. To test if a similar process occurred in WT cells, we performed the small vesicle generation assay with membranes from the 20K pellet fraction and cytosol collected from WT MEFs. As shown in *Figure 2—figure supplement 6*, small vesicles decorated with endogenous LC3 (both LC3-I and -II) were generated in a PI3K-COPII-dependent manner, whereas the packaging of COPII cargoes (SEC22B and ERGIC53) into small vesicles was COPII-dependent but PI3K-independent. Noticeably, small vesicles positive for the autophagic membrane protein ATG9 were also generated in the reaction, but in this case vesicles formed independent of PI3K or COPII (*Figure 2—figure supplement 6*), suggesting that this potential source of autophagosomal membrane was generated in a process distinct from that of the LC3 lipidation active vesicles. It is possible that the ATG9 vesicles come from the trans-Golgi network (TGN) as they have been shown to exit the TGN after starvation (*Young et al., 2006*). Due to technical limitations, it was difficult to distinguish cytosolic from peripheral vesicle-associated ATG proteins (data not shown).

To determine the requirement of COPII in autophagosome biogenesis, we depleted SEC23A by RNAi in WT MEFs and Hela cells (*Figure 2—figure supplement 5B, C*). Depletion of SEC23A reduced the biogenesis of autophagosomes as revealed by the formation of starvation-induced lipidated LC3 without or with the lysosome inhibitor bafilomycin A1 (*Figure 2—figure supplement 7A,B*) as well

as the reduced appearance of LC3 puncta in WT MEFs and Hela cells (*Figure 2—figure supplement 7C–F*). The data, together with the previous reports (*Hamasaki et al., 2003*; *Zoppino et al., 2010*; *Guo et al., 2012*; *Ge et al., 2013*; *Graef et al., 2013*; *Tan et al., 2013*), document a requirement for COPII in autophagosome biogenesis.

To further probe the relationship between COPII vesicles and LC3 lipidation, we examined the distribution of SEC31A, a key component of COPII (*Zanetti et al., 2012*), and endogenous LC3. Under nutrient-rich conditions, a few LC3 puncta formed which did not overlap with COPII vesicles (*Figure 2—figure supplement 8*). As expected, starvation induced a dramatic increase in LC3 puncta. On starvation, many small LC3 puncta colocalized with SEC31A whereas larger puncta did not (*Figure 2—figure supplement 8*).

The above data indicate that starvation activates PI3K in the ERGIC membrane followed by the action of COPII to generate vesicles that are active in lipidation of LC3. Under normal circumstances, COPII vesicles originate from ER-exit sites (ERES) and undergo at least partial uncoating before reaching the ERGIC (*Zanetti et al., 2012*; *Brandizzi and Barlowe, 2013*). Our finding of COPII vesicles budding from the ERGIC may be due to the starvation conditions used to induce autophagy. To test if COPII proteins are recruited to the ERGIC in starved cells, we incubated starved *Atg5* KO MEFs with or without wortmannin. We then isolated ERGIC membrane and immunoblotted fractions with antibodies against COPII proteins (*Figure 3A*). Without starvation, little COPII (SEC12, SAR1A and SEC23A; *Zanetti et al., 2012*) was detected on the ERGIC fraction (*Figure 3A*). Starvation increased the distribution of the COPII machinery, SEC12, SAR1A and SEC23A, to the ERGIC whereas inhibition of PI3K reduced this relocalization (*Figure 3A*). As controls, an ER resident protein, RPN1, and an ERGIC protein, SEC22B, remained at constant levels in the ERGIC fraction isolated from untreated and starved cells. As another control, localization of COPI (β-COP) on ERGIC membranes was not regulated by starvation or wortmannin (*Figure 3A*). We conclude that the conditions of starvation that induce autophagy result in the partial redistribution of COPII proteins and SEC12, the nucleotide exchange catalyst that activates SAR1 and initiates coat assembly, to the ERGIC membrane and that this change depends on active PI3K.

We examined the redistribution of COPII proteins to the ERGIC membrane by immunofluorescence confocal microscopy of normal and starved cells. ERGIC (ERGIC53) and COPII (SEC31A) appeared adjacent with marginal overlap in untreated cells (*Figure 3B,C* and *Figure 3—figure supplement 1*), consistent with previous work reporting a spatial relationship between the ERGIC and ERES where most of COPII coats reside (*Witte et al., 2011*). Starvation resulted in a qualitative and quantitative increase in the overlap between ERGIC and COPII markers (*Figure 3B,C*). This redistribution was reduced by inhibition of PI3K activity, consistent with the effect we observed with isolated membranes (*Figure 3A*). 3D reconstruction of the spatial distribution of ERGIC and COPII through deconvolution microscopy further supported the conclusion (*Video 1* and *Figure 3—figure supplement 2*), demonstrating the starvation-induced and PI3K-dependent recruitment of COPII to ERGIC.

Previous genetic and cell imaging studies of COPII as well as the structural analysis of TRAPPIII, a protein tethering complex essential for autophagy, have implied a close relationship between COPII vesicles and autophagosome formation (*Hamasaki et al., 2003*; *Hayashi-Nishino et al., 2009*; *Lynch-Day et al., 2010*; *Zoppino et al., 2010*; *Graef et al., 2013*; *Suzuki et al., 2013*; *Tan et al., 2013*). Here, we provide direct evidence demonstrating ERGIC-derived COPII vesicles, induced by the autophagic PI3K, serve as a membrane template for LC3 lipidation, a key step in autophagosome biogenesis (*Figure 3D*). Recent studies indicate the importance of LC3 lipidation in a late step involving the closure of the phagophore membrane to complete the autophagosome (*Sou et al., 2008*; *Kishi-Itakura et al., 2014*). Nonetheless, LC3 conjugation was also detected on the small phagophore membranes in the absence of VMP1, an early autophagic factor (*Kishi-Itakura et al., 2014*). Therefore LC3 lipidation occurs in an early stage of phagophore development even though it may function later. We propose that LC3-lipidated COPII vesicles may fuse homotypically or with membrane derived from other organelles such as the ER, plasma membrane, the ATG9 compartment and mitochondria, to facilitate maturation of the phagophare as well as completion of the double-membrane autophagosome (*Figure 3D*).

Other groups have localized autophagosome biogenesis to a phagophore assembly site (PAS) subdomain of ER (*Axe et al., 2008*; *Hayashi-Nishino et al., 2009*; *Yla-Anttila et al., 2009*; *Hamasaki et al., 2013a*). The COPII vesicles we have described may contribute membrane to the PAS. ATG14 has been shown to colocalize with LC3 on the phagophore membrane (*Itakura et al., 2008*; *Sun et al., 2008*;

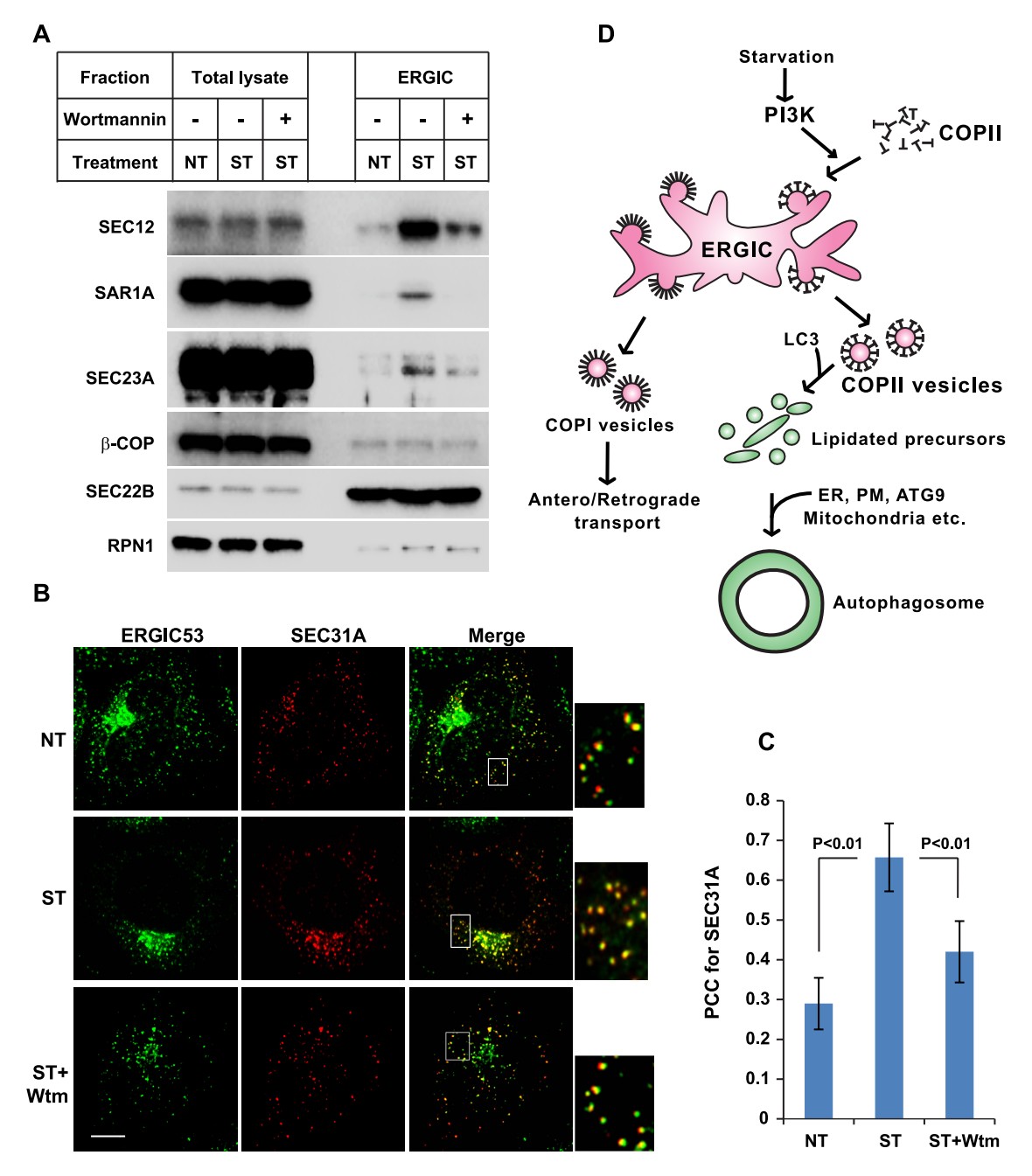

**Figure 3**. PI3K-dependent recruitment of COPII to ERGIC after starvation. (**A**) *Atg5* KO MEFs were either untreated or starved with or without 20 nM wortmannin for 30 min. Cells were harvested and ERGIC membranes were isolated by pooling fractions 3 and 4 (F2 in *Figure 2C,D*) of the OptiPrep gradient in the three-step fractionation approach followed by immunoblot to examine the amount of indicated markers on ERGIC membranes. (**B**) *Atg5* KO MEFs were transfected with a plasmid encoding an HA-tagged SEC31A. After transfection (24 hr), the cells were treated as shown in (**A**). Immunofluorescence was performed with anti-ERGIC53 and anti-HA antibodies and the cells were examined by confocal micros-copy. Bar: 10 μM (**C**) Quantification of SEC31A overlapped with ERGIC53 shown in (**B**) using the Pearson's correlation coefficient (PCC). Error bars represent standard deviations from ~20 cells in three independent experiments. p values were calculated by T-test. (**D**) A proposed model. Starvation activates autophagic PI3K to induce COPII recruitment on ERGIC. COPII vesicles generated from ERGIC serve as templates for efficient LC3 lipidation. These membrane precursors may collaborate with other membranes such as the endoplasmic reticulum (ER), plasma membrane (PM), ATG9 compartment and mitochondria, to form a mature autophagosome.

*Figure 3. Continued on next page*

*Figure 3. Continued*

The following figure supplements are available for figure 3:

**Figure supplement 1**. Extra images of *Figure 3B*.

**Figure supplement 2**. Histogram of colocalization between ERGIC53 and SEC31A in the deconvoluted 3D image.

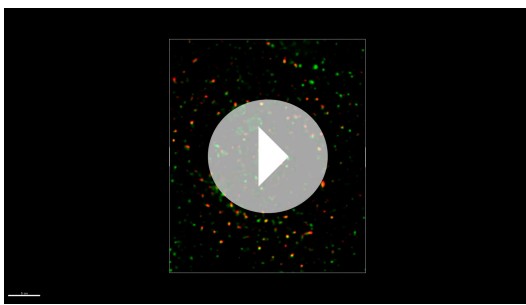

**Video 1**. 3D deconvolution images showing the distribution of ERGIC and COPII under conditions of untreated, starved or starvation with the PI3K inhibitor. *Atg5* KO MEFs were transfected with a plasmid encoding an HA-tagged SEC31A. Immunofluorescence labeling of ERGIC53 (green) and SEC31A (red) was performed as shown in *Figure 3B*. Z-stack images were collected and deconvolution was performed. 3D images were generated using Imaris imaging software. Cells with conditions of no treatment, starvation and starvation with 20 nM wortmannin were sequentially displayed.

*Matsunaga et al., 2009*, *2010*; *Zhong et al., 2009*). We showed that ATG14 and DFCP1 (a PI3P effector, *Axe et al., 2008*) are recruited to the ERGIC in starved cells (*Ge et al., 2013*). Therefore, it is possible that ATG14 may remain associated with and direct LC3-lipidated COPII vesicles to the PAS. ERGIC-localized ATG14 may be directly packaged into COPII vesicles active in LC3 lipidation. This association may be reinforced by the high curvature of COPII vesicles (60–80 nm; *Kim et al., 2005*; *Stagg et al., 2008*; *Zeuschner et al., 2006*), which are therefore a preferred binding site for the ATG14 protein (*Fan et al., 2011*). Alternatively, cytosolic ATG14 may be recruited to COPII vesicles after the budding process.

In summary, our data demonstrate a crosstalk between the autophagic PI3K and the COPII machinery leading to the generation of small vesicles derived from the ERGIC membrane that are active as a template in the lipidation of LC3. We suggest that starvation induces the movement of a fraction of SEC12, the key membrane protein required to initiate COPII assembly (*Zanetti et al., 2012*), from the ER to the ERGIC and that this, coupled to the activation of PI3K, permits the formation of specialized COPII vesicles that serve as the template for lipidation of LC3 and may further feed the growth of the phagophore.

## Materials and methods

### Materials, antibodies, plasmids and cell culture

Wortmannin, 3-methyladenine, brefeldin A, T7-LC3, GST, SAR1A (H79G), GST-FYVE and GST-FYVE(C/S) were as previously described (*Ge et al., 2013*). Bafilomycin A1 was purchased from LC Laboratories (Woburn, MA). Control siRNA was as described previously (*Ge et al., 2013*) and other siRNAs were purchase from Qiagen (Germantown, MD). SiRNAs used were mouse Sec23A (CAGTATTAATATAATGTTTAA, ACGGATGATGTTAGCTTACAA, AAGGATCTGTCTGCCAAACAA and CTCAGTTTATGTTTCATTTAA), human Sec23A (CAGACTCATAATAATATGTAT, CAGAGCCGGTTCTTCTTGATA, CACTACAACCTTAGCCATATA, ATGACGGTTGTAACTACTAAA), mouse FIP200 (CTGGAACAACTTGAAGAACAA, TAGGAACAATAAATTTATTAA, TTCATTGTATATTAACATTTA, CGGCTGGTAAATGAACAGAAA), mouse Atg14 (CTCCATCATATTCCCAATCGA, CCCGTGGATTAGCCTACCAAA, AGGACCTGACATGGAGCATAA, CACATACTTGACATCAATCTT) and mouse Beclin-1 (TTGGTTTGGAAAGATGCTTTA, CGGACAGTTTGGCACAATCAA, CGGGAGTATAGTGAGTTTAAA, TTGGGTAATATTAAACCACAT). The transfection of the siRNA was performed with Lipofectamine RNAiMAX (Invitrogen, Grand Island, NY).

Mouse anti-GM130, transferrin receptor, T7 and Tubulin; rabbit anti-SEC22B, Ribophorin 1, ERGIC53, SEC23A, SAR1A, LC3 and Beclin-1 antibodies were described before (*Ge et al., 2008*; *Schindler and Schekman, 2009*; *Ge et al., 2011*). We purchased mouse anti-HA from Covance (Emeryville, CA), mouse anti-LC3 (for immunofluorescence) and rabbit anti-ATG14 from MBL (Woburn, MA), rabbit anti-HA from Cell Signaling (Danvers, MA), rabbit anti-ATG9 from Novus Biologicals

(Littleton, CO), rabbit anti-FIP200 from Proteintech (Chicago, IL) and rabbit anti-β-COP from Abcam (Cambridge, MA). The plasmid encoding the HA-tagged human SEC31A was described before (*Jin et al., 2012*).

Atg5 KO and control MEFs (*Kuma et al., 2004*) were kindly provided by Noboru Mizushima. Reagents and procedures for cell culture were described previously (*Ge et al., 2013*).

## Cell-free small vesicle formation assay

For the collection of medium-speed membrane pellet, *Atg5* KO MEFs and WT MEFs were cultured, collected and lysed in B88 buffer as described previously (*Ge et al., 2013*). The lysate was centrifuged at 20K ×*g* for 10 min and the pellet fraction was washed once by suspending in B88 buffer (10 times volume of the pellet) followed by another centrifugation at 20K ×*g* for 10 min. For vesicle budding from the lysate of Sec23A knockdown cells, the 20K ×*g* membrane fraction was washed with 1.5 M urea dissolved in B88 buffer (10 times volume of the pellet) followed by a 20K ×*g* centrifugation for 10 min. The membrane fraction was further washed with B88 (10 times volume of the pellet) and centrifuged at 20K ×*g* for 10 min. The final membrane pellet was suspended in B88 lysis buffer (*Ge et al., 2013*) and the OD600 was adjusted to 10. For membrane fractionation, a three-step fractionation approach was performed (*Ge et al., 2013*). The indicated membrane fractions were collected by centrifugation and membranes were suspended in B88 lysis buffer to a concentration of 1 mg/ml phosphotidylcholine. *Atg5* KO MEF and WT MEF cytosols were prepared as previously described (*Ge et al., 2013*).

For each budding reaction, 5 µl membrane (OD600 = 10 for total membrane; 1 mg/ml phosphotidylcholine for membrane fractionation), 25 µl *Atg5* KO MEF cytosol (5 mg/ml), 0.75 µl GTP, 5 µl 10× ATP regeneration (*Ge et al., 2013*) and indicated drugs or proteins were incubated (GST-tagged protein purification was performed as described previously, *Ge et al., 2013*). B88 was added last to adjust to a final volume of 50 µl. The reaction was performed at 30°C for 1 hr followed by centrifugation at 20K ×*g* (for total membrane) or 25K ×*g* (for membrane fractionation) for 20 min. Supernatant aliquots (35 µl) were transferred to an ultracentrifuge tube for sedimentation at 55K rpm in a Beckman TLA100.3 rotor for 30 min. The supernatant fractions were removed and the small membranes were suspended in a 15 µl mixture containing starved HEK293T cytosol (2 mg/ml), GTP, an ATP regeneration mixture and 0.1 µg T7-LC3 (*Ge et al., 2013*) and incubated at 30°C for another 1 hr followed by SDS-PAGE and immunoblot to detect T7-LC3 lipidation.

## LC3 lipidation, membrane fractionation, immunodepletion, and immunoblot

These were performed as previously described (*Ge et al., 2008*, *2011*, *2013*).

## Immunofluorescence microscopy and quantification

Immunofluorescence was performed as previously described (*Ge et al., 2008*, *2011*). Confocal images were acquired with a Zeiss LSM 710 laser confocal scanning microscope (Molecular Imaging Center, UC, Berkeley). Colocalization of the confocal images was calculated by a pixel-based method by ImageJ (*Nakamura et al., 2007*). For deconvolution, image stacks were collected using a DeltaVision Elite microscope and deconvolution was performed by Huygens Professional 4.5.1p3 software. 3D video and the colocalization histogram of the deconvolution images were generated using Imaris 7.7.1 software (CNR, Biological Imaging Facility, UC, Berkeley). Quantification of the area of LC3 puncta was performed through Analyze Particles function of ImageJ. In brief, a threshold was set to calculate the total area of LC3 puncta which was divided by the area of the cell in the same image. The number was displayed as percentage of LC3 puncta area to the cell area.

## Acknowledgements

We thank Noboru Mizushima (University of Tokyo), Beth Levine (UT Southwestern Medical Center), James Hurley (UC Berkeley), Roberto Zoncu (UC Berkeley), Susan Ferro-Novick (UC San Diego) and Li Yu (Tsinghua University) for reagents, helpful discussion and advice, Steven Ruzin and Amita Gorur (UC Berkeley) for guidance on deconvolution, Bob Lesch, Ann Fischer and David Melville (UC Berkeley) for technical assistance. LG is supported by a fellowship from the Jane Coffin Childs Fund (JCCF). MZ is a research associate of HHMI. RS is an Investigator of the HHMI and a Senior Fellow of the UC Berkeley Miller Institute.

# Additional information

### Competing interests

RS: Editor in Chief, *eLife*. The other authors declare that no competing interests exist.

### Funding

| Funder | Author |
|---|---|
| Howard Hughes Medical Institute | Randy Schekman |
| Jane Coffin Childs Memorial Fund for Medical Research | Liang Ge |

The funders had no role in study design, data collection and interpretation, or the decision to submit the work for publication.

### Author contributions

LG, Conception and design, Acquisition of data, Analysis and interpretation of data, Drafting or revising the article, Contributed unpublished essential data or reagents; MZ, Acquisition of data, Drafting or revising the article, Contributed unpublished essential data or reagents; RS, Conception and design, Analysis and interpretation of data, Drafting or revising the article

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
