## [Decision Letter]

Thank you for sending your Research advance entitled “Phosphatidylinositol 3-kinase and COPII generate autophagosomal precursors from the ER-Golgi intermediate compartment” for consideration at *eLife*. Your article has been favorably evaluated by Tony Hunter (Senior editor) and 3 reviewers, one of whom, Noboru Mizushima, is a member of our Board of Reviewing Editors.

The Reviewing editor and the other reviewers discussed their comments before we reached this decision, and the Reviewing editor has assembled the following comments to help you prepare a revised submission.

In the previous report, Ge et al. showed that the ERGIC, which is enriched in the 25KP (25K pellet) membrane fraction, has the highest LC3 lipidation activity. In this follow-up study, the authors demonstrate that the 100KP membrane fraction also has an LC3 lipidation activity. They reveal that lipidation-active small vesicles enriched in the 100KP membrane fraction are generated from the 20KP membranes in a PI3K- and COPII-dependent manner. Finally, they show that COPII components are recruited to the ERGIC during starvation, again in a PI3K-dependent manner. These data suggest that PI3K and COPII are directly involved in generation of LC3 lipidation-active membranes from the ERGIC. This study will advance our understanding of the mechanisms of autophagosome formation.

Specific comments:

1) Although the data presented in the manuscript clearly suggest that the 100KP membranes are able to host LC3 lipidation, it is unclear whether these membranes are indeed incorporated in proceeding stages of biogenesis of autophagosomes. The authors should moderate their conclusion, for example in the Abstract, as their data do not support the reconstitution of autophagosome precursors but rather lipidation of LC3.

2) Related to the above criticism, it is important to test at which step endogenous LC3 and other ATGs are incorporated. In their previous paper, they showed colocalization of ERGIC with ATG14 and DFCP1 (12). However, this seems not to be the case in the model in the present manuscript. The localization of endogenous LC3-II on COPII vesicles both under starvation and non-starvation conditions should be determined. In addition, if the small vesicle generation assay is performed using wild-type 20KP, does the resulting 100KP contain LC3 and other ATGs? It is also recommended to include an LC3 blot in Figure 3.

3) In Figure 2, it is clearly shown that lipidation-active 100KP membranes are generated from the 20KP membranes. However, there is no direct evidence that they are indeed derived from the ERGIC. Can the authors deplete the ERGIC membranes from the 25KP membranes as performed in the previous study (12) and perform the lipidation assay?

4) Although the requirement for COPII in autophagosome formation *in vivo* has been suggested in yeast and also in the authors' previous study using the SAR1 mutants, the authors' hypothesis would be strengthened if the authors tested whether starvation-induced autophagy is indeed affected in COPII knock-down cells (e.g. in Sec23 RNAi cells in Figure 2).

5) Figure 3 should be performed more than n=1 (20 cells were counted from how many independent experiments?) and statistics should be performed.

6) In the Abstract (Figure 1, Figure 1) authors say the data are significant. They should provide a statistical analysis to support this.

7) In order to conclude that “autophagic PI3K” induces ERGIC-derived COPII vesicles, it is essential to determine the requirement for Atg14. Otherwise, this should be modified or simply stated as “PI3K”.

8) The authors discuss that LC3-positive COPII vesicles may fuse homotypically or with other membranes to “complete” the formation of autophagosomes. They could be correct that LC3-lipidation is important for the sealing step rather than the elongation step because neither the Atg12 nor Atg8/LC3 system is required for the membrane elongation (Sou et al. Mol Biol Cell. 2008,19:4762, Kishi-Itakura et al. J Cell Sci. 2014, Epub ahead of print). However, as the word “complete” is a little ambiguous, the authors may want to rephrase this sentence.

[Editors’ note: further clarifications were requested before acceptance, as described below.]

Thank you for resubmitting your Research advance entitled “Phosphatidylinositol 3-kinase and COPII generate LC3 lipidation vesicles from the ER-Golgi intermediate compartment” for further consideration at *eLife*. Your revised article has been favorably evaluated by Tony Hunter (Senior Editor) and a member of the Board of Reviewing Editors.

The authors have done an excellent, more than acceptable job of addressing the previous requests/criticisms. Although this manuscript is now acceptable, we would like to make a small suggestion that could be addressed before publication.

Regarding major point #2, the authors previously showed that ERGIC colocalizes with ATG14 and DFCP1 (12), whereas the present manuscript suggests that LC3 can be recruited to small vesicles derived from ERGIC. It implies that ATG14 and LC3 do not colocalize with each other. However, many papers have reported that ATG14 colocalizes with LC3 on phagosphores/isolation membranes (Itakura et al. MBC 2008, Matsunaga et al. NCB 2009, Matsunaga et al. JCB 2010, Zhong et al. NCB 2009). It would be helpful for readers if the authors could discuss this apparent discrepancy. Is ATG14 also incorporated into the P100K small vesicles? In addition, it is not clear in Figure 2—figure supplement 6, why the authors cannot test the presence of other ATGs (peripheral proteins) including ATG14 on the small vesicles. Some comment on this would be also helpful. If this experiment is feasible, the authors would be able to provide a clear explanation. The data showing that the small vesicle generation assay also produces Atg9 vesicles are very interesting.

---

## [Author Response]

*1) Although the data presented in the manuscript clearly suggest that the 100KP membranes are able to host LC3 lipidation, it is unclear whether these membranes are indeed incorporated in proceeding stages of biogenesis of autophagosomes. The authors should moderate their conclusion, for example in the Abstract, as their data do not support the reconstitution of autophagosome precursors but rather lipidation of LC3*.

We thank the reviewers for the suggestion. Modifications have been made accordingly.

*2) Related to the above criticism, it is important to test at which step endogenous LC3 and other ATGs are incorporated*.

We thank the reviewers for the suggestion. We performed three experiments as described in detail to answer each specific question below (Figure 2—figure supplement 3, Figure 2—figure supplement 6 and Figure 2—figure supplement 8). In brief, (1) the generation of LC3 lipidation-active vesicles is dependent on FIP200, ATG14 and Beclin-1, indicating this step could be downstream of the signaling pathway carried out through the FIP200/ULK1 complex and the autophagic PI3K complex (Figure 2—figure supplement 3); (2) The production of LC3 lipidation active vesicles employs a different pathway from those of the ATG9 vesicles, and these two population of vesicles could be generated in parallel in response to autophagic signals (Figure 2—figure supplement 6); (3) The COPII vesicles active in LC3 lipidation may act in an early stage of autophagosome biogenesis as small LC3-positive vesicles instead of big ones colocalize with COPII markers (Figure 2—figure supplement 8). Our future studies will employ the *in vitro* assays we have developed to reveal details in the mechanism of generation of these vesicles as well as the downstream fusion events needed to complete a double-membrane autophagosome.

*In their previous paper, they showed colocalization of ERGIC with ATG14 and DFCP1 (*[12]*). However, this seems not to be the case in the model in the present manuscript*.

Our current model is in line with our previous study. We showed that ATG14 and DFCP1 colocalized with the ERGIC (12) indicating the association of PI3K with the ERGIC as well as the production of autophagy-related PI3P on it. This result together with our current finding that the generation of LC3 lipidation active vesicles is dependent on PI3K and PI3P suggests a model wherein the recruitment of ATG14 together with the PI3K complex to the ERGIC produces PI3P, which is required for the direction of COPII machineries to the ERGIC for the generation of LC3 lipidation active vesicles.

*The localization of endogenous LC3-II on COPII vesicles both under starvation and non-starvation conditions should be determined*.

We thank the reviewers for the suggestion. We did the colocalization study of COPII and endogenous LC3 under normal and starved conditions as shown in Figure 2—figure supplement 8 in the revised manuscript. In nutrient-rich conditions, a few LC3 puncta are seen which do not overlap with SEC31A, a COPII component. Starvation induced the dramatic increase of LC3 puncta indicating the stimulation of autophagosome biogenesis. Interestingly, a fraction of small LC3 puncta colocalize with SEC31A, whereas bigger LC3 puncta were separated from SEC31A. The data confirm that COPII vesicles template the lipidation of LC3. Moreover, it also suggests that COPII vesicles may need to become at least partially uncoated to expose the membrane to the LC3 lipidation enzymes. Indeed, bigger LC3 puncta did not overlap with COPII.

In addition, if the small vesicle generation assay is performed using wild-type 20KP, does the resulting 100KP contain LC3 and other ATGs?

We thank the reviewers for the suggestion. We performed the suggested experiments as shown in Figure 2—figure supplement 6 in the current version of the manuscript. We observed the appearance of LC3-I and -II in the 100K fraction, dependent on PI3K and COPII. As our current budding assay only applies to the determination of integral membrane proteins or luminal proteins, we analyzed the distribution of ATG9, the only transmembrane ATG, in the 100K membrane fraction as shown in the same figure. The appearance of ATG9 in the 100K fraction was not dependent on either the PI3K or COPII, indicating that ATG9-containing vesicles are produced independently. As discussed in the manuscript, these ATG9 vesicles may originate from the trans-Golgi network where the formation of vesicles depends on other protein machineries.

*It is also recommended to include an LC3 blot in*
Figure 3.

We performed the LC3 blot. However, we did not detect any LC3 signal in the membrane fraction (data not shown) possibly because in Atg5KO MEF LC3-I could not strongly associate with membrane. Therefore it dissociated from the membrane during the three-step fractionation.

*3) In*
Figure 2*, it is clearly shown that lipidation-active 100KP membranes are generated from the 20KP membranes. However, there is no direct evidence that they are indeed derived from the ERGIC. Can the authors deplete the ERGIC membranes from the 25KP membranes as performed in the previous study (*[12]*) and perform the lipidation assay?*

We thank the reviewers for the suggestion. We performed an immunodepletion experiment as shown in Figure 2—figure supplement 4 in the revised manuscript. Removal of ERGIC by SEC22B antibody depletion abolished the generation of small vesicles active in LC3 lipidation, hence confirming the ERGIC as the membrane to generate these vesicles.

*4) Although the requirement for COPII in autophagosome formation in vivo has been suggested in yeast and also in the authors' previous study using the SAR1 mutants, the authors' hypothesis would be strengthened if the authors tested whether starvation-induced autophagy is indeed affected in COPII knock-down cells (e.g. in Sec23 RNAi cells in*
Figure 2*)*.

We thank the reviewers for the suggestion. We did Sec23A RNAi experiments as shown in Figure 2—figure supplement 7 (the effect of Sec23A knockdown on autophagosome biogenesis) and Figure 2—figure supplement 5, C (RNAi efficiency) in the revised manuscript. Depletion of SEC23A in WT MEF and Hela cells decreased starvation-induced LC3 lipidation as well as LC3 puncta formation. Therefore, COPII vesicles are involved in the biogenesis of the autophagosome.

*5)*
Figure 3
*should be performed more than n=1 (20 cells were counted from how many independent experiments?) and statistics should be performed*.

We thank the reviewers for the suggestion. We have indicated the number of experiments in the figure legends and T-tests were performed for statistics.

*6) In the Abstract (*Figure 1*,*
Figure 1*) authors say the data are significant. They should provide a statistical analysis to support this*.

We thank the reviewers for the suggestion. These data are representative of at least three experiments. Instead of showing statistical data to reconcile the word “significant”, we have employed a quantitative description in the current manuscript, which we think better describes the data.

*7) In order to conclude that “autophagic PI3K” induces ERGIC-derived COPII vesicles, it is essential to determine the requirement for Atg14. Otherwise, this should be modified or simply stated as “PI3K”*.

We thank the reviewers for the suggestion. We performed RNAi experiments to deplete FIP200 (one essential upstream factor) as well as ATG14 and Beclin-1 (key components of the autophagic PI3K complex) individually in Atg5KO MEF. As shown in Figure 2—figure supplement 3, cytosol collected from the autophagic factor-depleted cells displayed compromised activity to generate LC3 lipidation-active vesicles in the small vesicle generation assay, indicating the requirement of these upstream autophagic factors. Therefore, the data support the involvement of autophagic factors such as FIP200 and components of the autophagic PI3K in the induction of ERGIC-derived COPII vesicles. For the purpose of accurate description, we have removed “autophagic” from “autophagic PI3K” before Figure 2—figure supplement 3 in the text. However, we kept “autophagic” that appears after Figure 2—figure supplement 3, as this figure indicates the role of autophagic PI3K as well as upstream factors in this process.

*8) The authors discuss that LC3-positive COPII vesicles may fuse homotypically or with other membranes to “complete” the formation of autophagosomes. They could be correct that LC3-lipidation is important for the sealing step rather than the elongation step because neither the Atg12 nor Atg8/LC3 system is required for the membrane elongation (Sou et al. Mol Biol Cell. 2008,19:4762, Kishi-Itakura et al. J Cell Sci. 2014, Epub ahead of print). However, as the word “complete” is a little ambiguous, the authors may want to rephrase this sentence*.

We thank the reviewers for the suggestion. Modifications have been made and the references have been added.

*[Editors’ note: further clarifications were requested before acceptance, as described below*.*]*

*Regarding major point #2, the authors previously showed that ERGIC colocalizes with ATG14 and DFCP1 (*[12]*), whereas the present manuscript suggests that LC3 can be recruited to small vesicles derived from ERGIC. It implies that ATG14 and LC3 do not colocalize with each other. However, many papers have reported that ATG14 colocalizes with LC3 on phagosphores/isolation membranes (Itakura et al. MBC 2008, Matsunaga et al. NCB 2009, Matsunaga et al. JCB 2010, Zhong et al. NCB 2009). It would be helpful for readers if the authors could discuss this apparent discrepancy. Is ATG14 also incorporated into the P100K small vesicles? In addition, it is not clear in*
Figure 2—figure supplement 6*, why the authors cannot test the presence of other ATGs (peripheral proteins) including ATG14 on the small vesicles. Some comment on this would be also helpful. If this experiment is feasible, the authors would be able to provide a clear explanation. The data showing that the small vesicle generation assay also produces Atg9 vesicles are very interesting*.

We thank the reviewers for the suggestion. Our conclusions may be complementary rather than contradictory to those of other researchers. Our studies indicate that the ERGIC-localized PI3K functions to generate small vesicles as one template for LC3 lipidation and other researchers reported the colocalization of ATG14 with LC3-decorated phagophore. These data suggest a sequential role of ATG14 in autophagosome formation wherein it promotes the generation of LC3 lipidation-active vesicles from the ERGIC after which it may remain on these vesicles to provide one essential membrane for phagophore growth. COPII vesicles are small (60-80 nm) and highly curved and are thus ideal membrane targets for ATG14, which favors highly curved membrane (9). ERGIC-derived ATG14 may be packaged into COPII vesicles and remain associated through the process of lipidation and carried forward to the creation of the PAS by the fusion of vesicles including those that carry ATG9.

Alternatively, cytosolic ATG14 may be recruited to LC3 lipidation-active or LC3-lipidated vesicles after budding from the ERGIC. Either possibility requires the determination of ATG14 in the 100 KP COPII vesicle fraction. Unfortunately, our current assay could not reliably determine the level of ATG14 in the COPII fraction as cytosolic ATG14 and most of the ATGs tested sedimented in the 100k pellet fraction, which complicated our assessment of the origin of this pool of ATG proteins. This determination requires further effort, which we feel goes beyond the scope of our Advance report. We have included the above information in the revised manuscript as suggested by the reviewers.